# RealUnify: Do Unified Models Truly Benefit from Unification? A Comprehensive Benchmark

## Abstract

The integration of visual understanding and generation into unified multimodal models represents a significant stride toward general-purpose AI. However, a fundamental question remains unanswered by existing benchmarks: does this architectural unification actually enable *synergetic interaction* between the constituent capabilities? Existing evaluation paradigms, which primarily assess understanding and generation in isolation, are insufficient for determining whether a unified model can leverage its understanding to enhance its generation, or use generative simulation to facilitate deeper comprehension. To address this critical gap, we introduce **RealUnify**, a benchmark specifically designed to evaluate bidirectional capability synergy. RealUnify comprises 1,000 meticulously human-annotated instances spanning 10 categories and 32 subtasks. It is structured around two core axes: **1) Understanding Enhances Generation**, which requires reasoning (e.g., commonsense, logic) to guide image generation, and **2) Generation Enhances Understanding**, which necessitates mental simulation or reconstruction (e.g., of transformed or disordered visual inputs) to solve reasoning tasks. A key contribution is our **dual-evaluation protocol**, which combines direct end-to-end assessment with a diagnostic stepwise evaluation that decomposes tasks into distinct understanding and generation phases. This protocol allows us to precisely discern whether performance bottlenecks stem from deficiencies in core abilities or from a failure to integrate them. Through large-scale evaluations of 12 leading unified models and 6 specialized baselines, we find that current unified models still struggle to achieve effective synergy, indicating that architectural unification alone is insufficient. These results highlight the need for new training strategies and inductive biases to fully unlock the potential of unified modeling.

## 1 Introduction

The field of multimodal artificial intelligence has undergone a paradigm shift with the rise of unified models that integrate both visual understanding (e.g., visual question answering) and generation (e.g., text-to-image synthesis) within a single neural architecture (Deng et al., 2025; Lin et al., 2025; OpenAI, 2025). While such unification offers architectural elegance, its most compelling promise lies in the potential for synergetic effects between capabilities: leveraging knowledge and reasoning from understanding to guide more accurate *generation*, and employing internal generative simulation (e.g., "thinking with images") to facilitate more complex *understanding*. A fundamental open question remains: do the two core capabilities, i.e., understanding and generation, mutually enhance each other? This question further motivates a reconsideration of architectural design: should we pursue a unified model that integrates both, or co-located models without functional synergy?

A primary obstacle in answering this question is *the lack of a suitable benchmark*. As illustrated in Figure 1, current evaluation frameworks predominantly assess understanding and generation in isolation (Stage 1), and some benchmarks (Xie et al., 2025b) combine tasks from both domains to evaluate capabilities simultaneously (Stage 1.5). Recent efforts like WISE (Niu et al., 2025) have begun exploring whether understanding enhances generation quality, but do not explicitly test whether success on a task *depends* on the interaction of both capabilities. Thus, there remains a pronounced lack of rigorous benchmarks with systematic design to probe the very essence of unification: bidirectional capability synergy. For example, while text-to-image benchmarks excel at measuring output fidelity and aesthetic quality, they cannot determine whether a model's reasoning

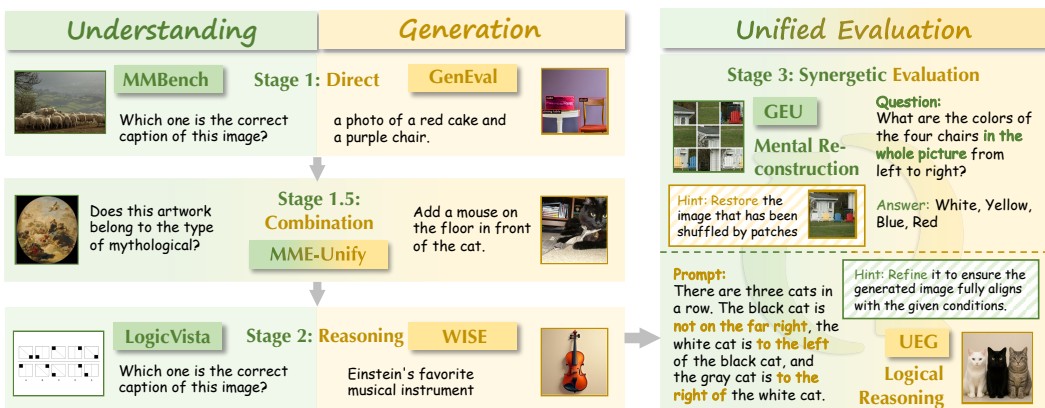

Figure 1: **Illustration of RealUnify**. Unlike benchmarks focused on either understanding or generation (Stage 1), those that merely integrate both capabilities (Stage 1.5), or even those that preliminarily explore the mutual enhancement between understanding and generation (Stage 2), RealUnify stands as the first benchmark to comprehensively evaluate and fully harness the synergy between these capabilities, making it a pioneering effort in assessing ability synergy for unified models.

abilities meaningfully enhance its generative process, nor whether generative simulation improves visual understanding.

To address this gap, we introduce RealUnify, the first benchmark aimed at answering the fundamental question: Can unified models effectively leverage their synergy between understanding and generation abilities to solve complex tasks? The core innovation of RealUnify lies in its meticulously designed task suite, where each instance requires an intricate interplay between understanding and generation. RealUnify structures its evaluation around two core tracks: (1) *Understanding Enhances Generation* (UEG), which tests whether knowledge and reasoning improve generation accuracy, and (2) *Generation Enhances Understanding* (GEU), which examines whether generative reconstruction and visualization can support more effective visual reasoning. As shown in Table 1, its $1,000$ instances span $10$ categories and $32$ manually crafted and validated subtasks, which in particular requires synergy between understanding and generation, such as tasks that require mathematical computation prior (understanding) to generate images and visual tracking of multi-step transformations (generation) to answer questions (understanding). Moreover, a cornerstone of RealUnify is its *dual-evaluation protocol* including the direct evaluation and stepwise evaluation, enabling precise diagnosis of whether unified models achieve genuine capability synergy or merely functional coexistence. Specifically, direct evaluation tests whether models can achieve end-to-end synergy in a realistic setting (closer to the intrinsic capability of models during the realistic deployment), whereas stepwise evaluation decomposes tasks into understanding and generation, revealing whether performance limits arise from weak individual capabilities or from the lack of genuine synergy.

Through extensive evaluations of $12$ leading unified models and $6$ state-of-the-art specialized baselines using our dual-evaluation protocol, we uncover a striking conclusion: despite their unified architecture, current models still struggle to synergize understanding and generation capabilities effectively. This finding is robustly supported by three key empirical patterns. First, under *direct evaluation*, models perform poorly on both UEG (average $37.5\%$ for best open-source) and GEU tasks, indicating their inability to spontaneously integrate capabilities in end-to-end scenarios. Second, and more diagnostically, the *stepwise evaluation* reveals a revealing dissociation: when UEG tasks are decomposed into "understanding-then-generation" stages, performance improves significantly (e.g., BAGEL (Deng et al., 2025) improves from $32.7\%$ to $47.7\%$), demonstrating that models possess the required knowledge but cannot seamlessly integrate it. Conversely, decomposing GEU tasks into "generation-then-understanding" stages causes performance to degrade, suggesting that models default to relying on understanding shortcuts rather than effectively leveraging generation. Third, when we construct an "oracle" model by combining the best specialist models (Gemini-2.5-Pro (Comanici et al., 2025) for understanding and GPT-Image-1 (OpenAI, 2025) for generation) in a stepwise manner, it achieves $72.7\%$ on UEG tasks, establishing a high-performance upper bound that current unified models fall far short of. Collectively, these results indicate that architectural unification alone is insufficient. To fully realize the potential of capability synergy, unified models require more advanced training schemes and stronger inductive biases.

## 2 RELATED WORK

**Unified Multimodal Models.** Recently, unified models (Wang et al., 2024a; Chen et al., 2025b; Team, 2024; Deng et al., 2025; Huang et al., 2025; Wu et al., 2025b; Wang et al., 2025a; Lin et al., 2025) have emerged as a central research direction in multimodal intelligence. These frameworks integrate both visual understanding and generation within a single architecture and have demonstrated competitive performance. Early studies (Chen et al., 2025b; Wang et al., 2024a; Team, 2024) primarily emphasize functional integration, ensuring that a single model could simultaneously perform understanding and generation tasks. With the evolution of model capabilities, research interest has shifted toward examining whether unification itself can yield additional benefits or even give rise to emergent abilities. For example, Liquid (Wu et al., 2024) provides empirical evidence that training data from either understanding or generation tasks can enhance performance on the other, indicating reciprocal benefits between the two. Building on this finding, UniFluid (Fan et al., 2025) shows that well-designed training strategies can further reinforce such cross-task gains. Beyond performance improvements, BAGEL (Deng et al., 2025) uncovers the emergence of complex compositional behaviors, such as multimodal generation with long-context reasoning. Extending this perspective, Doracycle (Zhao et al., 2025) introduces cyclic paradigms—text-to-image-to-text and image-to-text-to-image—that enable self-evolution of unified models without reliance on annotated data.

Table 1: **Comparisons on RealUnify and other benchmarks.** RealUnify is designed to provide a comprehensive evaluation of unified models across multiple dimensions. It is entirely human-annotated and integrates both direct and stepwise evaluation protocols. A distinctive feature of RealUnify is its focus on assessing whether the synergy between generation and understanding can be effectively harnessed to solve complex tasks.

| Benchmark | Category | #QA | #Tasks | #Subtasks | Annotation | Eval | Und. | Gen. | Ability Synergy |
|---|---|---|---|---|---|---|---|---|---|
| MME-Unify (Xie et al., 2025b) | Unified | 4,104 | 14 | - | Mixed | Direct | ✓ | ✓ | ✗ |
| UniEval (Li et al., 2025d) | Unified | 1,234 | 13 | 81 | (M)LLM | Direct | ✓ | ✓ | ✗ |
| T2I-CoReBench (Li et al., 2025c) | T2I | 1,080 | 2 | 12 | (M)LLM | Direct | ✗ | ✓ | ✗ |
| Science-T2I (Li et al., 2025b) | T2I | 898 | 3 | 16 | Human | Direct | ✗ | ✓ | ✗ |
| T2I-ReasonBench (Sun et al., 2025) | T2I | 800 | 4 | 26 | (M)LLM | Direct | ✗ | ✓ | ✗ |
| WISE (Niu et al., 2025) | T2I | 1,000 | 3 | 25 | Mixed | Direct | ✗ | ✓ | ✗ |
| MMBench (Liu et al., 2024) | I2T | 3,217 | 6 | 20 | Mixed | Direct | ✓ | ✗ | ✗ |
| LogicVista (Xiao et al., 2024) | I2T | 448 | 5 | 9 | Human | Direct | ✓ | ✗ | ✗ |
| **RealUnify** | Unified | 1,000 | 10 | 32 | Human | Direct/Step | ✓ | ✓ | ✓ |

**Benchmarks for Unified Models.** Research on unified models has recently emerged, driving the need for benchmarks tailored to their evaluation. Among existing efforts, MME-Unify (Xie et al., 2025b) is the first benchmark to jointly assess multimodal comprehension, generation, and mixed-modality tasks. Building on this direction, UniEval (Li et al., 2025d) enables evaluation without auxiliary models or human annotations. Despite these advances, these benchmarks still fall short of assessing whether integrating understanding and generation actually produces measurable performance gains. Complementary to these works, several text-to-image (T2I) benchmarks, such as MMMG (Luo et al., 2025), T2I-CoReBench (Li et al., 2025c), and WISE (Niu et al., 2025), can also be adapted for evaluating unified models, but their emphasis on T2I tasks provides limited evidence of reciprocal benefits between understanding and generation. To address this gap, we introduce RealUnify, a benchmark explicitly designed to test whether unification enhances both capabilities in a synergetic way, offering a more comprehensive and reliable framework for evaluating unified models.

## 3 REALUNIFY

The tasks in RealUnify fall into two categories: assessing whether model understanding enhances generation (UEG) (Section 3.1), and whether generative ability supports understanding (GEU) (Section 3.2). As shown in Figure 2, these categories jointly evaluate the extent to which cross-capability transfer improves complex task performance and overall model competence. Details on dataset construction and evaluation are provided in Section 3.3.

### 3.1 UNDERSTANDING ENHANCES GENERATION (UEG)

For the UEG tasks, we focus on a thorough evaluation of the image generation capabilities of current unified models. To emphasize the role of understanding in the image generation process, we design 6 categories in which the model must first interpret the prompt before producing the output.

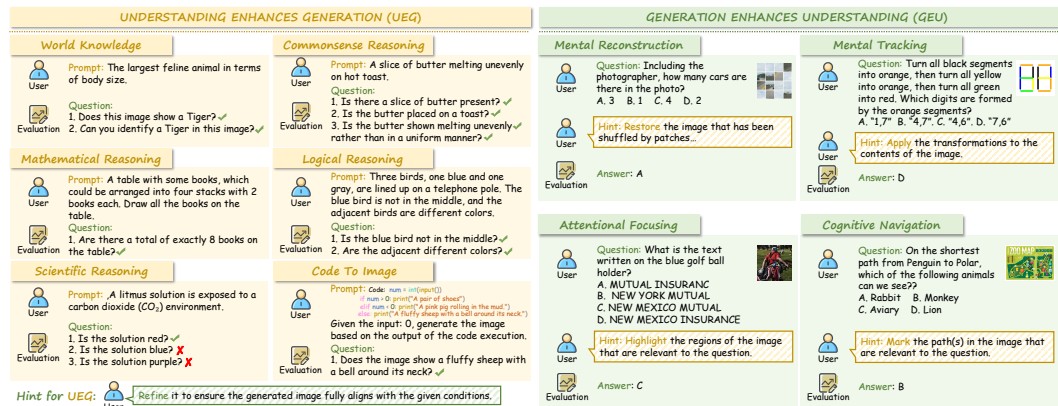

Figure 2: **Overview of RealUnify.** The benchmark includes 2 main task categories: Understanding Enhances Generation (UEG) and Generation Enhances Understanding (GEU), encompassing 10 distinct task types. Hints are provided to guide task decomposition in the stepwise evaluation.

**World Knowledge.** This task category targets image generation grounded in objective world knowledge. The goal is to examine whether unified models can accurately produce visual content that aligns with established facts. To ensure comprehensive coverage, the tasks span 7 major domains of knowledge: *Animals & Plants*, *Food*, *Architecture*, *Culture*, *Sports*, *Technology*, and *Lifestyle*, enabling a broad assessment of models' ability to leverage world knowledge across diverse scenarios.

**Commonsense Reasoning.** Commonsense reasoning requires the model to generate images that reflect everyday phenomena observed in the real world. In this category, the model is given prompts describing widely recognized real-world situations, and it should produce corresponding images. The evaluation focuses on whether models demonstrate commonsense intelligence, such as understanding basic physical laws, human activities, and common objects in daily life.

**Mathematical Reasoning.** Mathematical reasoning and computation represent fundamental yet essential abilities for intelligent systems. In this category, models are required to perform the necessary calculations implied by the image-generation instructions in order to produce correct results. The tasks cover a range of operations, including *Numerical Calculation (Single-Step & Multi-Step)*, *Probability Estimation*, *Proportional Reasoning*, and *Constraint-based Equation Solving*.

**Logical Reasoning.** Logical reasoning is a cornerstone of intelligence, enabling models to trace dependencies, combine multiple conditions into consistent outcomes, and adapt their outputs to hypothetical changes. This category assesses whether models can reason over explicit or implicit conditions to ensure that generated outputs satisfy logical constraints.

**Scientific Reasoning.** Tasks in this category require reasoning grounded in specialized scientific principles across 4 domains: *Physics*, *Chemistry*, *Biology*, and *Geography*. The goal is to assess whether models can correctly apply established scientific principles to reason about given scenarios and generate outputs consistent with real-world phenomena.

**Code-to-Image.** This category evaluates the model's ability to bridge symbolic code and visual generation. Specifically, the model must parse the provided code, reason over its logic in conjunction with the given input, and infer the correct textual instruction implied by the execution outcome. It is then required to generate an image that faithfully reflects this inferred instruction.

## 3.2 GENERATION ENHANCES UNDERSTANDING (GEU)

For the GEU tasks, we focus on questions that require the model to leverage its generative capabilities to simplify problem-solving and thereby improve overall accuracy. To this end, we design 4 customized tasks that evaluate the model's understanding capability.

**Mental Reconstruction.** This task type evaluates a model's ability to reason over and reconstruct disrupted visual inputs. Images are divided into patches of varying granularity and then shuffled. The model is required to answer specific questions based on the disordered image, accurate responses

typically hinge on the ability to consider spatial arrangement, matching, or relational relations. Success in these tasks necessitates accurate reconstruction of the original image.

**Mental Tracking.** This task aims to test the model's ability to trace and update visual states through a sequence of transformations. The input consists of digits constructed from colored line segments, and the model is instructed to perform diverse modifications—for example, "`first changing all blue segments to green, then turning all green segments into yellow`", and so on. The model is then queried about the digit represented by a specific color. Successfully completing such tasks requires the model to internally track and memorize how different regions evolve under successive changes.

**Attentional Focusing.** This category examines whether unified models can effectively concentrate on critical regions within complex visual inputs, a paradigm often associated with the notion of "thinking with images" (Zhang et al., 2025; Su et al., 2025). Common techniques for emphasizing salient content include cropping, bounding-box annotation, or super-resolution. For unified models, however, a key challenge is whether they can leverage their native visual generation ability to highlight target regions directly, thereby facilitating more accurate visual understanding. This task category encompasses 3 sub-tasks, involving *Quantity Recognition*, *OCR Recognition*, and *Attribute Recognition*, which collectively assess a model's capacity to extract and reason over essential information.

**Cognitive Navigation.** Navigation is an essential task in real-world scenarios, where models must proceed step by step to ultimately reach a defined goal. In this category, we distinguish between two types of navigation tasks. The first type is *maze navigation*, in which we synthesize multiple mazes and require models to answer questions that involve solving them. The second type is *map navigation*, which we consider more representative of real-world conditions. Beyond simply solving the map, this task evaluates the model's ability to identify the shortest route or follow specific paths under given constraints, thereby providing a more comprehensive assessment of its navigation capabilities.

## 3.3 BENCHMARK CONSTRUCTION

In this section, we first describe the sources and collection process of our data. We then present the two evaluation settings of RealUnify, followed by a summary of its key statistics.

**Data Collection and Annotation.** We collect data from multiple sources. For the UEG tasks, all prompts are manually curated by 10 human experts. After collection, we perform a cross-check in which three additional reviewers independently validate the prompts, and only those agreed upon by all reviewers are retained. For the GEU tasks, we develop an automated script to generate samples for Mental Reconstruction and Mental Tracking tasks, which are then annotated by human experts. A cross-checking procedure is similarly applied to ensure data correctness. For the Attentional Focusing task, we sample data from BLINK (Fu et al., 2024) and HR-Bench (Wang et al., 2025b). For the Cognitive Navigation task, mazes are generated automatically, with human experts providing the corresponding answers. Additionally, map images are sourced via the Google Search API, and the associated questions and answers are created by human experts.

**Evaluation Criteria.** To further investigate whether unified models can truly benefit from unification, we design two complementary evaluation protocols: direct evaluation and stepwise evaluation. Direct evaluation focuses on measuring the overall performance of unified models, examining whether unification leads to notable gains in an end-to-end manner. In contrast, stepwise evaluation explicitly decomposes each task into separate stages of understanding and generation, allowing a fine-grained analysis of model strengths and weaknesses and providing clearer evidence of whether unification contributes to improved capability in solving complex tasks.

**Direct Evaluation.** In direct evaluation, the model is required to perform the tasks described in Sections 3.1 and 3.2 without intermediate decomposition. For the Understanding Enhances Generation setting, tasks take the form of text-to-image generation, where the model must directly produce a target image given a problem statement. For the Generation Enhances Understanding setting, tasks adopt an image-to-text format, in which the model is provided with an image, a question, and multiple candidate options, and is expected to select the most appropriate option. As is shown in Figure 3, to verify the correctness of the generated images in the text-to-image setting, we further employ a question list to poll the outputs, ensuring that the visual content aligns with the intended target.

**Stepwise Evaluation.** In stepwise evaluation, tasks in RealUnify are explicitly decomposed into 2 sequential stages. For the Understanding Enhances Generation setting, the unified model must first solve the problem in pure text form and then use the obtained response as the instruction for subsequent image generation, which is consistent with the "first understanding, then generation" paradigm widely adopted in text-to-image models (Deng et al., 2025; Guo et al., 2025; Liao et al., 2025). In contrast, for the Generation Enhances Understanding setting, the model is required to first produce an intermediate image based on the given input and then answer the corresponding question using this image, which is consistent with the "first generation, then understanding" strategy commonly explored in works on "thinking with images" (Su et al., 2025; Yang et al., 2025; Zhang et al., 2025). This protocol not only enables a finer-grained analysis of potential bottlenecks in unified models but also provides clearer evidence of whether unification leads to genuine performance gains.

**Benchmark Statistics.** After a rigorous process of question construction, selection, and subsequent annotation and verification by domain experts, we compile a dataset consisting of 1, 000 questions, with 600 UEG tasks and 400 GEU tasks. As illustrated in Figure 4, these questions cover a wide range of categories, resulting in 32 distinct subtasks distributed in multiple domains. This dataset provides a systematic framework for evaluating the capabilities of unified models in a synergetic manner, offering insight into their effectiveness in complex real-world tasks.

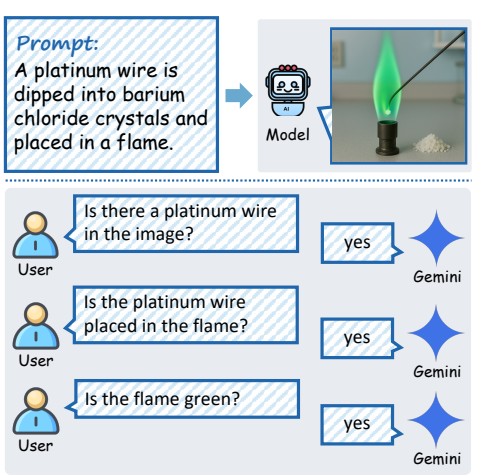

Figure 3: **Illustration of polling evaluation.** To assess the accuracy of the generated images, we meticulously design verification questions and employ Gemini 2.5 Pro as the judge in a polling-based evaluation.

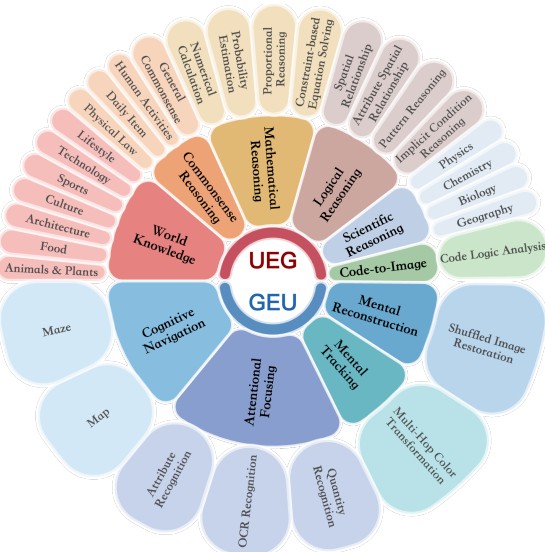

Figure 4: **Statistics of RealUnify.** The tasks span 10 categories, divided into two main groups: UEG and GEU, including 32 subtasks or areas.

## 4 EXPERIMENTS

We conduct extensive experiments with RealUnify across various models to evaluate both existing unified frameworks and prior state-of-the-art models for understanding and generation. Section 4.1 describes our experimental setup, followed by the main results of RealUnify. In Section 4.2, we provide additional analyses to illustrate the challenges posed by our benchmark and present case studies that highlight the gap between current unified models and truly synergetic unified frameworks.

### 4.1 EVALUATION ON REALUNIFY

**Evaluation Setup.** For the evaluation on RealUnify, we consider a total of 12 unified models: 11 state-of-the-art open-source models and 1 cutting-edge closed-source model. Among the open-source models, we select representative candidates, including BAGEL-7B (Deng et al., 2025), OmniGen2 (Wu et al., 2025b), Ovis-U1-3B (Wang et al., 2025a), UniWorld-V1 (Lin et al., 2025), UniPic2-Metaquery-9B (Wei et al., 2025), OneCAT-3B (Li et al., 2025a), MIO (Wang et al., 2024b),

ILLUME+ (Huang et al., 2025), Show-o2 (Xie et al., 2025a), Janus-Pro (Chen et al., 2025b), and BLIP3-o (Chen et al., 2025a). In addition, we evaluate the closed-source Gemini-2.5-Flash-Image (Google, 2025), also referred to as "Nano Banana", which serves as a strong baseline.

To better quantify the performance gap between unified and state-of-the-art specialized models in visual understanding and generation, we evaluate 3 high-performing models from each domain. For visual understanding, we include Gemini-2.5-Pro (Comanici et al., 2025), GPT-4.1 (Achiam et al., 2023), and Qwen2.5-VL-7B (Bai et al., 2025). For image generation, we assess FLUX.1 Kontext (Batifol et al., 2025), Qwen-Image (Wu et al., 2025a), and GPT-Image-1 (OpenAI, 2025). Taken together, these evaluation results provide a comprehensive overview of the capabilities of current unified models as well as cutting-edge understanding and generation models on RealUnify.

**Main Results.** We evaluate all unified models on RealUnify using both direct and stepwise evaluation, as described in Section 3.3. The results for each task category, as well as the overall performance, are summarized in Table 2. All performance values are reported as accuracy percentages.

Table 2: **Evaluation results on RealUnify. WR**: World Knowledge; **CR**: Commonsense Reasoning; **MR-I**: Mathematical Reasoning; **LR**: Logical Reasoning; **SR**: Scientific Reasoning; **C2T**: Code-to-Image; **MR-II**: Mental Reconstruction; **MT**: Mental Tracking; **AF**: Attentional Focusing; **CN**: Cognitive Navigation. For each task, we present both direct and stepwise evaluation results, reported in the format direct/step. The best performance on each task is in blue .

| Model | Understanding Enhances Generation | | | | | | | Generation Enhances Understanding | | | | | Total |
|---|---|---|---|---|---|---|---|---|---|---|---|---|---|
| | WK | CR | MR-I | LR | SR | C2I | Avg | MR-II | MT | AF | CN | Avg | |
| *Proprietary Models* | | | | | | | | | | | | | |
| Nano Banana | 89 / - | 86 / - | 34 / - | 65 / - | 48 / - | 56 / - | 63.0 / - | 34 / - | 27 / - | 36 / - | 30 / - | 31.8 / - | 50.5 / - |
| *Open-Source Unified Models* | | | | | | | | | | | | | |
| MIO | 24 / 35 | 26 / 33 | 18 / 13 | 9 / 10 | 10 / 11 | 0 / 8 | 14.5 / 18.3 | 26 / 23 | 19 / 18 | 35 / 19 | 23 / 21 | 25.8 / 20.3 | 19.0 / 19.1 |
| Janus-Pro | 25 / 26 | 77 / 71 | 16 / 7 | 13 / 17 | 16 / 20 | 3 / 10 | 25.0 / 25.2 | 21 / - | 23 / - | 28 / - | 29 / - | 25.3 / - | 25.1 / - |
| ILLUME+ | 44 / 52 | 62 / 62 | 22 / 22 | 23 / 25 | 26 / 26 | 1 / 7 | 29.7 / 32.3 | 27 / 27 | 19 / 20 | 35 / 38 | 30 / 25 | 27.8 / 27.5 | 28.9 / 30.4 |
| Show-o2 | 30 / 42 | 56 / 50 | 25 / 25 | 21 / 21 | 18 / 20 | 18 / 19 | 28.0 / 29.5 | 36 / - | 28 / - | 36 / - | 21 / - | 30.3 / - | 28.9 / - |
| OmniGen2 | 36 / 55 | 61 / 60 | 21 / 26 | 29 / 28 | 16 / 20 | 19 / 6 | 30.3 / 32.5 | 30 / 42 | 21 / 24 | 51 / 38 | 28 / 19 | 32.5 / 30.8 | 31.2 / 31.8 |
| UniPic2 | 61 / 62 | 73 / 72 | 31 / 30 | 28 / 38 | 25 / 26 | 7 / 15 | 37.5 / 40.5 | 26 / 28 | 20 / 24 | 27 / 27 | 23 / 16 | 24.0 / 23.8 | 32.1 / 33.8 |
| UniWorld-V1 | 51 / 56 | 64 / 59 | 26 / 26 | 33 / 37 | 21 / 24 | 15 / 9 | 35.0 / 35.2 | 29 / 33 | 19 / 25 | 57 / 36 | 24 / 20 | 32.3 / 28.5 | 33.9 / 32.5 |
| Ovis-U1 | 37 / 59 | 72 / 71 | 28 / 30 | 23 / 34 | 15 / 17 | 12 / 25 | 31.2 / 39.3 | 32 / 38 | 28 / 25 | 60 / 31 | 36 / 24 | 39.0 / 29.5 | 34.3 / 35.4 |
| BLIP3-o | 57 / 62 | 71 / 74 | 21 / 24 | 19 / 25 | 28 / 22 | 2 / 9 | 33.0 / 36.0 | 36 / - | 25 / - | 57 / - | 32 / - | 37.5 / - | 34.8 / - |
| OneCAT | 61 / 64 | 70 / 65 | 32 / 20 | 29 / 27 | 24 / 31 | 9 / 27 | 37.5 / 39.0 | 26 / 29 | 25 / 26 | 43 / 26 | 31 / 36 | 31.3 / 29.3 | 35.0 / 35.1 |
| BAGEL | 46 / 74 | 70 / 80 | 23 / 26 | 29 / 37 | 21 / 29 | 7 / 40 | 32.7 / 47.7 | 37 / 38 | 31 / 25 | 50 / 52 | 39 / 28 | 39.3 / 35.8 | 35.3 / 42.9 |

As shown in Table 2, under direct evaluation on RealUnify, existing unified models perform poorly on both UEG and GEU tasks, underscoring the gap between current unified approaches and true task unification. In particular, UEG tasks reveal a marked performance disparity between open-source and proprietary models. While the best open-source model achieves 37.5, the proprietary Nano Banana (Google, 2025) reaches 63.0. This highlights the difficulty open-source unified models face in leveraging their understanding capabilities to support generation inherently. In contrast, the GEU tasks reveal a different pattern. Although all models still perform poorly, open-source models demonstrate notably stronger understanding capabilities than proprietary models. This further confirms our earlier conclusion. Despite their promising comprehension abilities, current unified models struggle to effectively incorporate such understanding into the generation process. Bridging this gap between understanding and generation is crucial for enhancing the performance of these models, particularly when dealing with complex generation tasks.

To further explore the capabilities of current unified models, we conduct experiments using the stepwise evaluation framework. Since certain models do not support image editing, their results on the GEU tasks are not reported. By decoupling both UEG and GEU tasks, our goal is to uncover the true potential of these models and to further analyze their stepwise performance on tasks that demand both understanding and generation abilities. The results reveal a quite surprising pattern. For UEG tasks, all models benefit from stepwise decomposition, with BAGEL showing the most substantial improvement (+15). These results suggest that current unified models can internally retain the knowledge required for complex generation tasks. However, they struggle to inherently leverage this knowledge in practice in the UEG tasks, indicating that they remain far from achieving genuine task unification. In the case of GEU tasks, the situation differs considerably. After stepwise decomposition, all models exhibit reduced performance. This outcome indicates that although current unified models possess adequate generative capabilities, they still lack the ability to effectively apply these capabilities

to real-world problem-solving. The observed degradation further suggests that, in direct evaluation, these models tend to rely primarily on their understanding abilities while overlooking the fact that GEU tasks demand a synergetic integration of both generation and understanding.

Taken together, these results suggest that although current unified models possess sufficient understanding and generation capabilities individually, they fall short on tasks that require a synergetic integration of both. This shortcoming highlights a persistent performance gap between existing approaches and the goal of achieving true unification.

Table 3: **Performance comparison of unified models and specialized models.** We report results by selecting the top-3 performing unified models based on their overall performance in UEG and GEU and comparing them against specialized models. The best performance on each task is in blue .

(a) Understanding Enhances Generation (UEG)

| Model | WK | CR | MR-I | LR | SR | C2I | Total |
|---|---|---|---|---|---|---|---|
| *Specialized Models* | | | | | | | |
| GPT-Image-1 | 90 | 87 | 31 | 69 | 48 | 48 | 62.2 |
| Qwen-Image | 66 | 83 | 28 | 44 | 25 | 67 | 52.2 |
| FLUX.1 Kontext | 53 | 73 | 25 | 27 | 25 | 37 | 40.0 |
| *Unified Models* | | | | | | | |
| Nano Banana | 89 | 86 | 34 | 65 | 48 | 56 | 63.0 |
| UniPic2 | 61 | 73 | 31 | 28 | 25 | 7 | 37.5 |
| OneCAT | 61 | 70 | 32 | 29 | 24 | 9 | 37.5 |

(b) Generation Enhances Understanding (GEU)

| Model | MR-II | MT | AF | CN | Total |
|---|---|---|---|---|---|
| *Specialized Models* | | | | | |
| Gemini 2.5 Pro | 30 | 73 | 73 | 43 | 54.8 |
| GPT-4.1 | 38 | 23 | 56 | 37 | 38.5 |
| Qwen2.5-VL | 35 | 23 | 44 | 36 | 34.5 |
| *Unified Models* | | | | | |
| BAGEL | 37 | 31 | 50 | 39 | 39.3 |
| Ovis-U1 | 32 | 28 | 60 | 36 | 39.0 |
| BLIP3-o | 36 | 25 | 57 | 32 | 37.5 |

## 4.2 ANALYSIS EXPERIMENTS

In this section, we provide a detailed analysis of the challenges presented by RealUnify. To investigate the limitations of current unified models, we conduct experiments to compare unified models with generation and understanding specialists. Additionally, we present a case study that illustrates in greater depth why these models fall short on RealUnify.

**Comparisons with Specialists.** We then compare the top 3 unified models, ranked by their overall performance on the UEG and GEU tasks, against the leading specialized models. As shown in Table 3, unified models demonstrate competitive results on UEG tasks, even surpassing state-of-the-art image generation models in some cases. This suggests that incorporating understanding capabilities can indeed facilitate complex generation tasks. Nevertheless, the comparison also reveals that, for challenging image generation tasks, current open-source models still exhibit a substantial performance gap relative to proprietary counterparts, indicating that further progress in model architecture, dataset construction, and large-scale training is required to close this gap. On GEU tasks, by contrast, open-source unified models already demonstrate strong understanding abilities, even outperforming certain proprietary specialist models. These findings underscore the value of synergetic unified frameworks, which not only exploit strong understanding capabilities to improve generation, narrowing the gap with proprietary generation models, but also integrate generative components to enrich understanding, making the overall process more intuitive and efficient.

**How well can unified models be?** To better examine the potential upper bound of current unified models, we conduct experiments by combining two of the most powerful models for generation and understanding: Gemini-2.5-Pro (Comanici et al., 2025) and GPT-Image-1 (OpenAI, 2025). This combination is evaluated in a stepwise manner to approximate the maximum performance that unified models could achieve in task unification.

As shown in Table 4, integrating these two strong models yields an impressive score of 72.7 on UEG tasks. This result not only demonstrates that our UEG tasks require substantial reasoning capabilities for successful generation, but also highlights that, although current unified models benefit from stepwise decomposition on UEG tasks, they remain far from achieving comparable performance in a truly synergetic fashion.

Table 4: **Comparisons with Gen-Und SOTA.**

| Model | WK | CR | MR-I | LR | SR | C2T | Total |
|---|---|---|---|---|---|---|---|
| Nano Banana | 89 | 86 | 34 | 65 | 48 | 56 | 63 |
| Und→Gen (SOTA) | 93 | 86 | 43 | 70 | 53 | 91 | 72.7 |

| Model | MR-II | MT | AF | CN | Total |
|---|---|---|---|---|---|
| BAGEL | 37 | 31 | 50 | 39 | 39.3 |
| Gen→Und (SOTA) | 29 | 27 | 21 | 50 | 31.8 |

For GEU tasks, the results further support our earlier conclusion that, although current state-of-the-art generative models can produce photorealistic images, they remain inadequate for addressing real-

world problems. When GPT-Image-1 (OpenAI, 2025) is integrated with Gemini-2.5-Pro (Comanici et al., 2025), we observe substantial performance degradation, underscoring that adapting generative capabilities for practical problem-solving remains a significant challenge for unified models. Strengthening the generalization capacity of generative models is thus a key step toward developing more reliable and effective unified frameworks.

**Case Study.** We further present qualitative results to illustrate how unified models address tasks and to highlight the performance gap relative to ideal settings. Figure 5 shows 2 examples revealing how Bagel (Deng et al., 2025) benefits from stepwise execution, which compels it to integrate both understanding and generative capabilities. For UEG tasks, by explicitly leveraging its understanding ability, the model correctly infers that the object required is a lightsaber and is then able to generate an appropriate image. For GEU tasks, although the model does not perfectly reconstruct the disordered image patches, the intermediate reconstruction still guides it toward the correct answer. These examples reveal how unified models can benefit from the synergy between understanding and generation, enabling them to solve problems that neither capability alone could address.

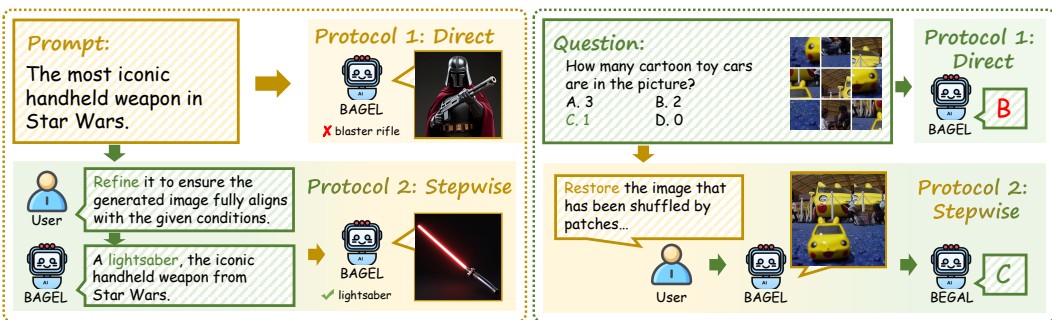

Figure 5: **Effective examples of stepwise execution in task solving.** Through the unified model's inherent understanding and generation abilities, the model is able to implement complex tasks.

We further examine the oracle settings and provide two illustrative examples to demonstrate their performance when supplied with ground-truth intermediate results. As shown in Figure 6, the Bagel model continues to underperform even under step-wise evaluation. However, the performance improves when the ground-truth intermediate step is given. This observation suggests that existing unified models still lack essential internal capabilities. Such limitations may hinder the effective integration of understanding and generation in real-world problem-solving, underscoring the need to strengthen the core capacities of unified models as a prerequisite for true unification.

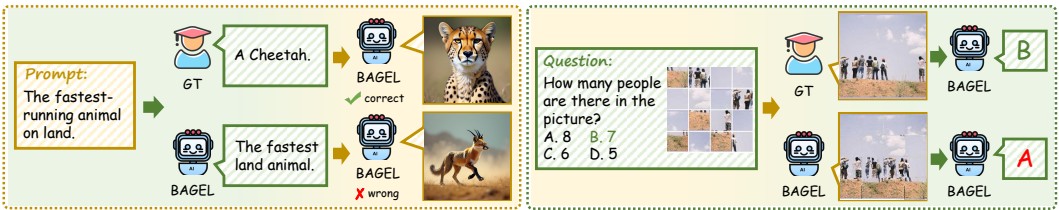

Figure 6: **Challenging examples of stepwise execution in task solving.** Despite using a stepwise approach, the unified model struggles to complete complex tasks, only succeeding with intermediate results based on the given ground truth.

## 5 CONCLUSION

In this paper, we introduce RealUnify, the first comprehensive benchmark explicitly designed to investigate capability synergy in unified models. It systematically evaluates this synergy through two complementary settings—Understanding Enhances Generation and Generation Enhances Understanding—across 10 diverse task categories. Extensive experiments and analysis reveal that current unified models are still far from achieving genuine synergy. Although they can accomplish tasks under stepwise decomposition, indicating substantial potential, their inability to succeed in end-to-end scenarios highlights the absence of true synergy. Realizing such synergy in unified models, and thus empowering them to tackle complex real-world tasks, remains a pressing research direction.

## ETHICS STATEMENT

This research strictly follows the ICLR Code of Ethics. We emphasize that the collected dataset is entirely non-commercial, with its development carefully avoiding any ethical or legal issues, particularly concerning intellectual property. Our methodology rigorously upholds copyright integrity through two protective measures: 1) all foundational project descriptions are the original creations of the authors and student volunteers; 2) we strictly adhere to the licenses of collected images, ensuring no legal issues arise from data collection or release.

## REPRODUCIBILITY STATEMENT

To ensure the reproducibility of our results, we have made considerable efforts to provide the necessary details and materials. Specifically, we have included a comprehensive description of the dataset creation process in Section 3. More implementation details and benchmark details are described in Appendix B and Appendix C.

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

## APPENDIX

We provide supplementary documents to support our research. The details of Large Language Model usage are presented in Section A. We provide visualization results of representative examples for each subtask of RealUnify, along with the overall task distribution in Section B. In addition, implementation details are outlined in Section C to enhance the reproducibility of our results. Finally, in Section D, we present common failure modes of unified models in generation tasks.

## A  LARGE LANGUAGE MODEL USAGE

In this paper, we clarify that Large Language Models (LLMs) are employed solely to support and refine the writing process. Specifically, we use LLMs to provide sentence-level suggestions and to enhance the overall fluency of the text.

## B  BENCHMARK DETAILS

### B.1  REPRESENTATIVE EXAMPLES FROM REALUNIFY

In order to comprehensively convey the characteristics of tasks in RealUnify, two representative examples are presented for each task. Figure 7, 8, 9, and 10 present examples of the Understanding Enhances Generation (UEG) tasks and Generation Enhances Understanding (GEU) tasks, respectively.

### B.2  TASK DISTRIBUTION

Table 5 presents the distribution of task instances across different categories in RealUnify. Each task is evaluated under both direct and stepwise settings. In the latter, the evaluation is decomposed into two parts: one focusing on visual understanding and the other on generation, thereby allowing a more fine-grained assessment of the model's reasoning process.

Table 5: **Distribution of task instances across different categories in RealUnify.** Each task is evaluated under both direct and stepwise settings, where stepwise evaluation further decomposes the process into a visual understanding problem and a generation problem.

| Task Category | Task | #Number (Direct / Stepwise) |
|---|---|---|
| Understanding Enhances Generation | World Knowledge | 100 / 100 |
| | Commonsense Reasoning | 100 / 100 |
| | Mathematical Reasoning | 100 / 100 |
| | Logical Reasoning | 100 / 100 |
| | Scientific Reasoning | 100 / 100 |
| | Code-to-Image | 100 / 100 |
| Generation Enhances Understanding | Mental Reconstruction | 100 / 100 |
| | Mental Tracking | 100 / 100 |
| | Attentional Focusing | 100 / 100 |
| | Cognitive Navigation | 100 / 100 |
| Total | - | 1,000 / 1,000 |

## C  EXPERIMENT DETAILS

### C.1  EVALUATION SETUP

We evaluate a total of 12 unified models on RealUnify, including 11 leading open-source models and 1 cutting-edge proprietary model.

For the proprietary model, we evaluate Gemini 2.5 Flash Image (also known as "Nano Banana") (Google, 2025) using the official API, `gemini-2.5-flash-image-preview`.

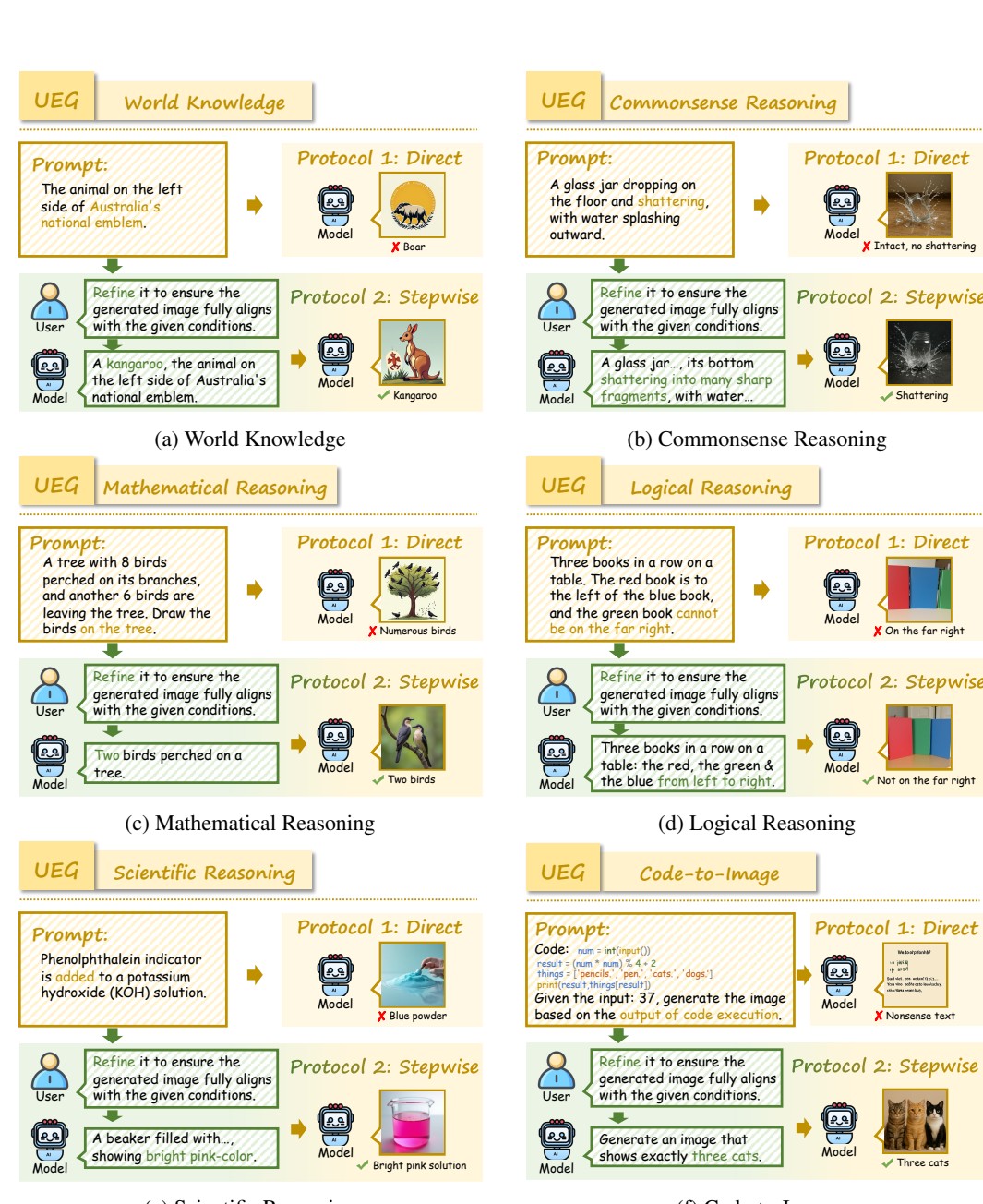

Figure 7: **Examples of Understanding Enhances Generation (UEG) tasks in RealUnify.**

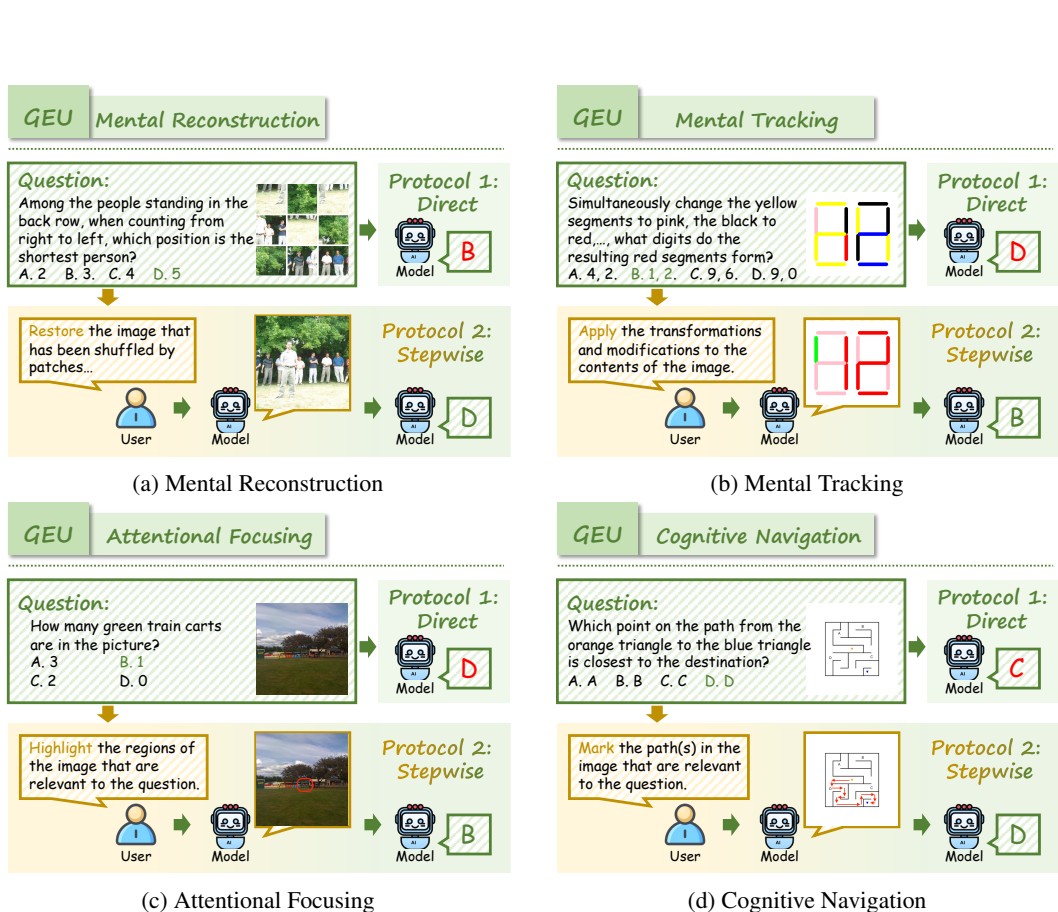

(a) Mental Reconstruction

(b) Mental Tracking

(c) Attentional Focusing

(d) Cognitive Navigation

Figure 8: **Examples of Generation Enhances Understanding (GEU) tasks in RealUnify.**

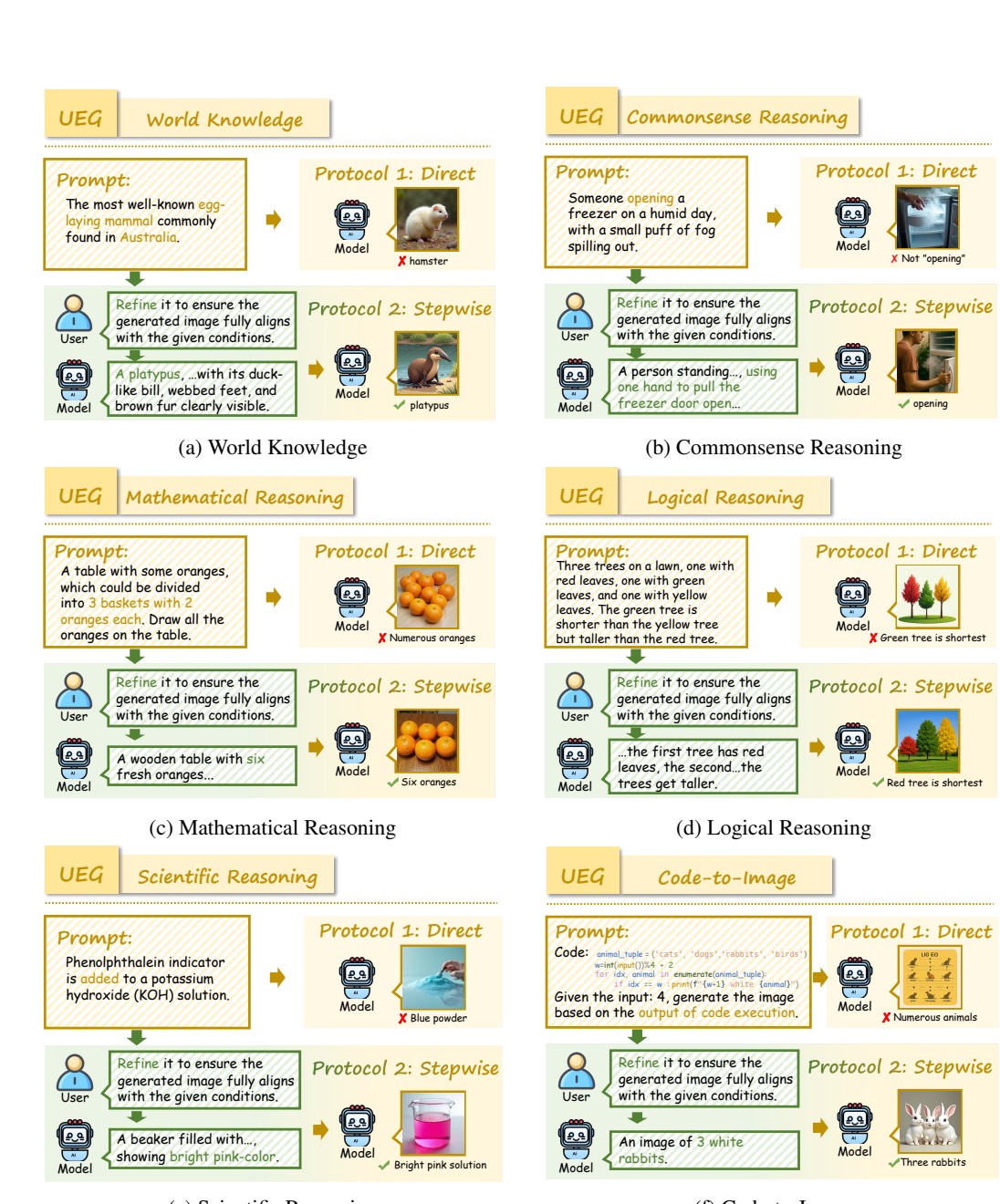

Figure 9: **Examples of Understanding Enhances Generation (UEG) tasks in RealUnify.**

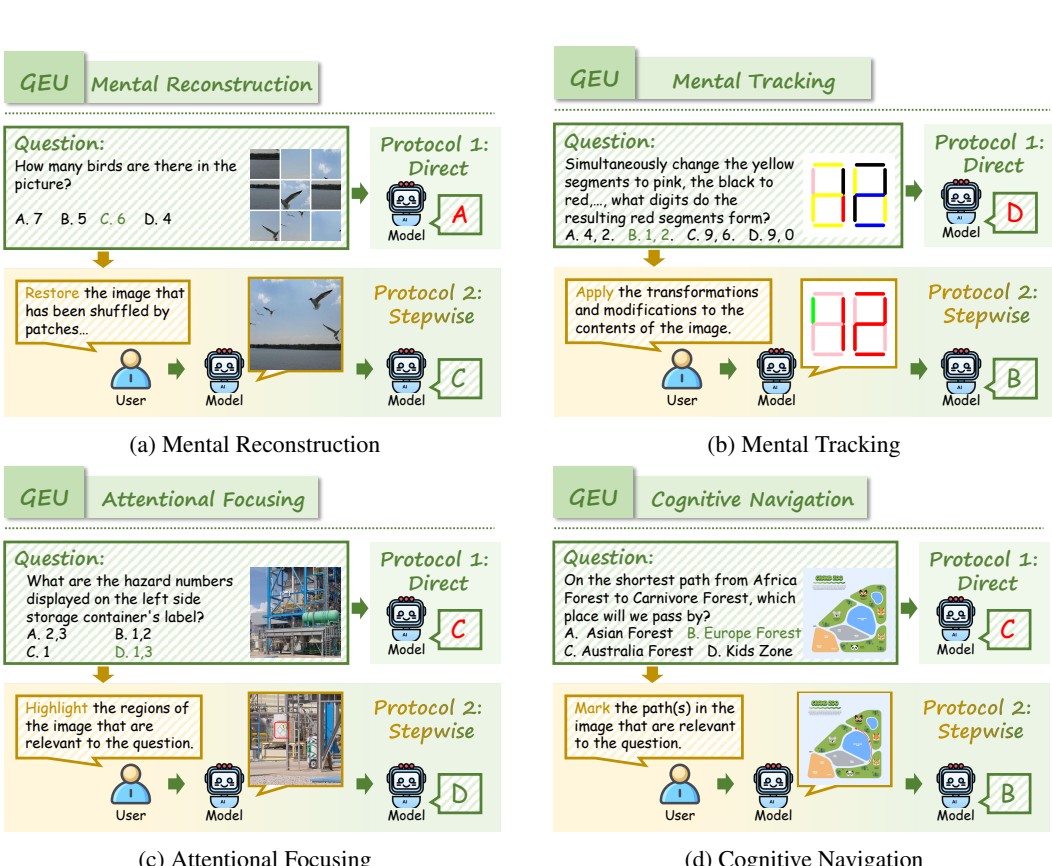

(a) Mental Reconstruction

(b) Mental Tracking

(c) Attentional Focusing

(d) Cognitive Navigation

Figure 10: **Examples of Generation Enhances Understanding (GEU) tasks in RealUnify.**

Table 6: **Polling prompt using Gemini 2.5 Pro as the judge model in UEG tasks.**

```
[Image]
```
Please answer the following question based on the image:
Question: `[Question]`

You should only reply yes or no, and do not provide any other extra content.

For open-source models, we select BAGEL-7B (Deng et al., 2025), OmniGen2 (Wu et al., 2025b), Ovis-U1-3B (Wang et al., 2025a), UniWorld-V1 (Lin et al., 2025), UniPic2-Metaquery-9B (Wei et al., 2025), OneCAT-3B (Li et al., 2025a), MIO (Wang et al., 2024b), ILLUME+ (Huang et al., 2025), Show-o2 (Xie et al., 2025a), Janus-Pro (Chen et al., 2025b), and BLIP3-o (Chen et al., 2025a). All models are evaluated using the official default or recommended settings for inference.

In the Understanding Ehances Generation (UEG) tasks, we use the state-of-the-art Gemini 2.5 Pro (Comanici et al., 2025) as the judge model to evaluate the generated images through a polling-based method. The evaluation is performed through the official `gemini-2.5-pro` API.

## C.2 EVALUATION PROMPT

For the Understanding Enhances Generation (UEG) tasks, when polling the generated images using Gemini 2.5 Pro (Comanici et al., 2025), we use the prompt provided in Table 6.

Table 7: **Evaluation prompt for the multiple-choice question in GEU tasks.**

```
[Image]
```
Select the best answer to the following multiple-option question based on the image. Respond with only the letter (A, B, C, or D) of the correct option.
Question: `[Question]`
Option:
A. `[Option A]`
B. `[Option B]`
C. `[Option C]`
D. `[Option D]`
The best answer is:

For the Generation Ehances Understanding (GEU) tasks, since the tasks are presented in the multiple-choice format, we provide the prompt for the multiple-choice questions in Table 7.

Table 8: **Prompt for Understanding Enhances Generation (UEG) tasks.**

Here is the prompt for image generation: `[Prompt]`

Please refine it into a simple, direct, and unambiguous form to ensure the generated image fully aligns with the given description and conditions.

Respond only with the refined prompt, without adding anything else.

In the stepwise evaluation of the Understanding Enhances Generation (UEG) tasks, the models first need to refine the original prompt. The corresponding prompt is provided in Table 8.

In the stepwise evaluation of the Generation Ehances Understanding (GEU) task, each task is decomposed, with image generation (editing) performed first, followed by visual understanding. Tables 9, 10, 11, and 12 present the prompts used for image generation (editing) in the Mental Reconstruction, Mental Tracking, Attentional Focusing, and Cognitive Navigation tasks, respectively.

Table 9: **Prompt for stepwise evaluation of Mental Reconstruction tasks.**

```
[Image]
```
Please restore the image that has been shuffled by patches, without adding extra content or altering the original image.

Table 10: **Prompt for stepwise evaluation of Mental Tracking tasks.**

```
[Image]
```
Here is the question: `[Question]`
Please apply the corresponding transformations and modifications to the contents of the image according to the question.

Table 11: **Prompt for stepwise evaluation of Attentional Focusing tasks.**

```
[Image]
```
Here is the question: `[Question]`
Please highlight the regions of the image that are relevant to the question.

Table 12: **Prompt for stepwise evaluation of Cognitive Navigation tasks.**

```
[Image]
```
Here is the question: `[Question]`
Please mark the path(s) in the image that are relevant to the question.

## D   COMMON FAILURE MODES OF UNIFIED MODELS IN GENERATION TASKS

Even state-of-the-art unified models still exhibit typical failure modes during image generation, including attribute entanglement, inaccurate quantity, attribute fidelity errors, and confused spatial relationships. We illustrate these common failure modes in Figure 11 and Figure 12.

As shown in Figure 11, when the instruction involves generating multiple objects or objects of different types with distinct attributes, the model often exhibits attribute mixing between different objects and mismatches in object quantity. In addition, when the objects to be generated have specific or complex attributes and structures, the model is also prone to insufficient fidelity. Moreover, the accurate realization of spatial relationships among multiple objects remains a common issue for the model. Figure 12 exposes several other problems of the model. First, in generating fine-grained features such as fingers and text, the model often suffers from detail loss, distortion, and deformation. Second, the model is also prone to generating scenes that violate common sense and physical laws. Finally, even for common and clearly defined objects (e.g., a lioness), the model shows severe confusion, such as generating features of a male lion instead. These errors reveal the clear shortcomings and typical failure modes of unified models in the generation process, limiting their performance on more complex tasks. In particular, for challenging tasks such as RealUnify, which require the synergy of multiple capabilities, these issues may become significant bottlenecks.

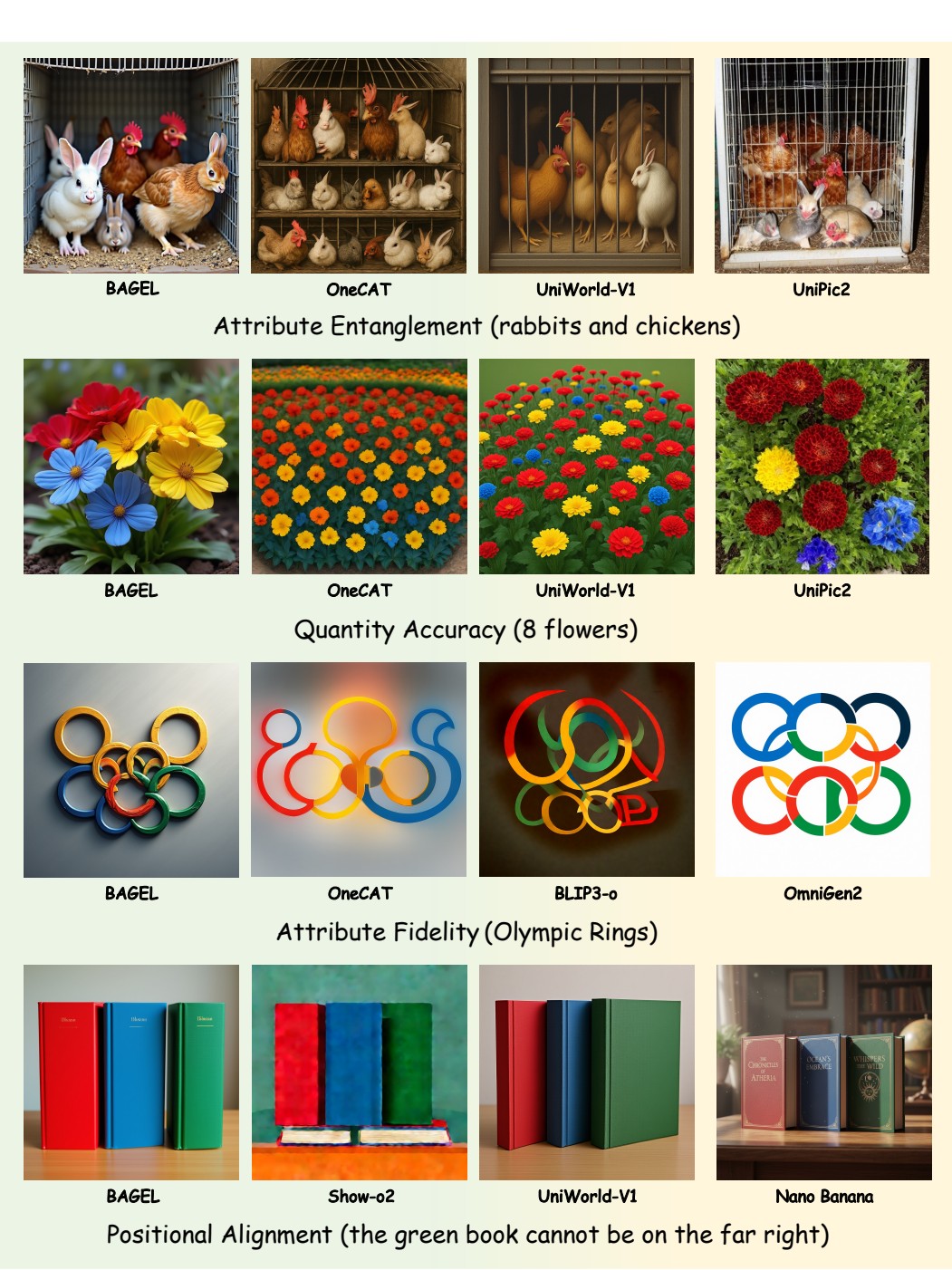

Figure 11: **Common failure modes of unified models during image generation.**

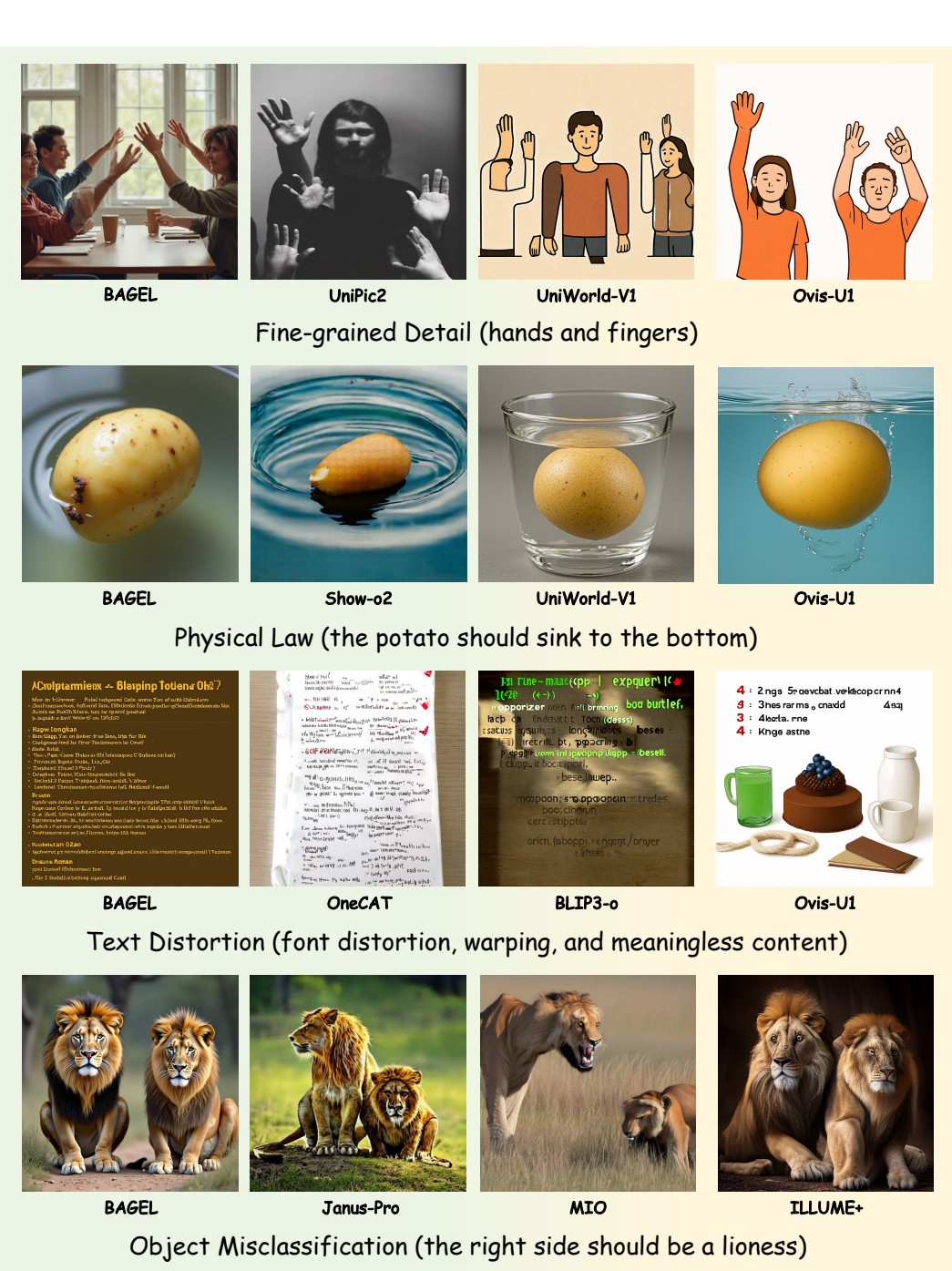

Figure 12: **Common failure modes of unified models during image generation.**

