# OpenReview forum: "RealUnify: Do Unified Models Truly Benefit from Unification? A Comprehensive Benchmark"
_ICLR.cc/2026/Conference — ICLR 2026 Conference Withdrawn Submission_

### Official Review · Reviewer_Cjtx · 2025-10-19

**Soundness:** 3
**Presentation:** 3
**Contribution:** 3
**Rating:** 6
**Confidence:** 5

**Summary:**

This paper introduce RealUnify, a benchmark with 1,000 human-annotated instances covering 10 categories, focusing on two core tasks: Understanding Enhances Generation (UEG) and Generation Enhances Understanding (GEU). RealUnify uses two evaluation methods—direct and stepwise—to find performance bottlenecks. They tested 12 unified models and 6 specialized ones, finding current unified models still lack effective synergy, so architectural unification alone is not enough.

**Strengths:**

1. This paper is generally well-written and easy to follow, with clearly illustrated figures.
2. The dual-evaluation protocol (direct + stepwise) helps clearly identify whether poor performance comes from weak individual abilities or lack of synergy, which is effective and reasonable.
3. It compares unified models with specialized ones and builds an "oracle" model (combining top specialists) to set a performance upper bound, making results more convincing.

**Weaknesses:**

1. The authors do not directly verify the necessity of "thinking with images" for GEU tasks, which weakens the rationality of their conclusion about "models overlooking synergy". This leaves room to doubt: some GEU tasks might be solvable with pure understanding, and the stepwise performance drop could stem from added task burden (not poor synergy), weakening the conclusion that "models rely on understanding shortcuts".
2. When evaluating image generation in UEG tasks, it only uses a question list to check semantic correctness (e.g., whether the generated content matches the prompt’s meaning) but ignores quality-related metrics, making the generation evaluation incomplete.

**Questions:**

Please see weaknesses.

---

### Official Review · Reviewer_tFKs · 2025-10-30

**Soundness:** 1
**Presentation:** 2
**Contribution:** 1
**Rating:** 2
**Confidence:** 4

**Summary:**

This paper introduces RealUnify, a human-annotated benchmark designed to evaluate whether unified multimodal models exhibit bidirectional synergy across visual understanding and image generation. The benchmark comprises 1,000 instances spanning 10 categories and 32 subtasks, organized into Understanding Enhances Generation (UEG) and Generation Enhances Understanding (GEU) tracks. The authors propose a dual evaluation protocol—direct end-to-end execution and stepwise decomposition—to diagnose whether performance bottlenecks stem from deficiencies in core capabilities or from failures to integrate them.Empirical results show that current unified models perform poorly in direct settings, suggesting that architectural unification alone does not yield synergy. An “oracle” constructed by chaining specialist models significantly outperforms unified models, highlighting unrealized potential and indicating the need for improved training strategies and inductive biases.

**Strengths:**

+The paper addresses a key limitation of prior benchmarks (e.g., MME-Unify, WISE) that only evaluate understanding/generation in isolation or simple combination, not bidirectional synergy—aligning with the core promise of unified models.

+The benchmark spans 10 categories, including world knowledge, commonsense, logic, mathematics, scientific reasoning, and code-to-image, plus four GEU tasks involving mental reconstruction/tracking, offering a broad lens on potential synergy.

+Evaluations include 12 unified models (e.g., BAGEL, Gemini-2.5-Flash-Image), 6 specialist models (e.g., Gemini-2.5-Pro, GPT-Image-1), and an oracle configuration, providing informative comparative context。

**Weaknesses:**

-Misaligned Evaluation: UEG tasks rely solely on textual instructions without multimodal inputs. As a result, UEG effectively reduces to instruction following followed by standard text-to-image prompting, rather than demonstrating cross-modal synergy. The pattern where the performance of stepwise > direct evaluation on UEG but stepwise < direct on GEU suggests the benchmark rewards pipeline decomposition more than emergent integrated reasoning. The strong oracle performance further indicates that improvements stem from chaining specialists rather than unification, weakening the synergy claim.

-Several GEU tasks can plausibly be solved by strong understanding models (e.g., o3-like systems) using tool use, or “think-with-image” capabilities without invoking genuine image editing or image-to-image generation. Conversely, some UEG instances can be solved via text-only prompt refinement. This blurs whether tasks force cross-ability dependence as intended.

-Dataset Opacity & Reproducibility Risks: Section 3.3 offers minimal annotation detail (lines 247–255): missing annotator backgrounds, recruitment and compensation, annotation guidelines, quality-control procedures, adjudication rules, inter-annotator agreement metrics, and dataset diversity analyses. Without these, dataset reliability cannot be verified and reproducibility is limited. For a human-annotated benchmark, this level of disclosure is insufficient.

-Interpretation of stepwise evaluation gains: Reported improvements under the stepwise setting (e.g., BAGEL UEG 32.7% → 47.7%) are attributed to “synergy failure.” However, they more likely reflect shortcomings in end-to-end instruction decomposition. The oracle’s 72.7% UEG accuracy suggests the benefits come from structured pipelines, not internal synergy.

-Limited novelty: RealUnify primarily complexifies existing tasks (e.g., adding multi-step reasoning to text-to-image pipelines or introducing patch-level reconstruction variants for image QA), rather than designing novel paradigms that inherently require synergistic interaction between multimodal understanding and generation. As a result, the benchmark does not fully evaluate whether models can benefit from both abilities simultaneously, nor does it isolate circumstances under which generation enhances understanding or vice versa. It remains unclear whether RealUnify measures synergy, pipeline competence, or general robustness to compositional instructions, limits its novelty.

**Questions:**

1. Dataset Reliability: Can you provide (1) annotator backgrounds and domain expertise, (2) Inter-annotator agreement metrics, (3) details on  disagreement protocols, (4) recruitment and payment, annotation guidelines for all human-labeled subsets, and (5) Dataset diversity statistics.

2.Multimodal Design in UEG: UEG prompts rely solely on text. Could you elaborate on the rationale behind excluding multimodal conditioning (e.g., reference images)? Would incorporating such inputs better capture cross-modal synergy rather than instruction following and prompt refinement?

3.Role of tool-calling on GEU: Have you evaluated models with tool-calling capabilities or think with images capabilities (e.g., O3-like) on GEU tasks? If these models can succeed on GEU tasks without explicit image generation, how should we interpret GEU’s validity as a synergy-driven metric?

---

### Official Review · Reviewer_d7bN · 2025-10-31

**Soundness:** 2
**Presentation:** 3
**Contribution:** 2
**Rating:** 4
**Confidence:** 3

**Summary:**

The paper introduces RealUnify, a benchmark for evaluating unified multimodal models through the bidirectional interaction between understanding and generation. It covers 10 task categories and 32 subtasks with two evaluation dimensions, Understanding Enhances Generation and Generation Enhances Understanding. The benchmark supports both direct and step-by-step evaluation to diagnose integration failures.

**Strengths:**

1. The paper presents a well-motivated and timely benchmark that explores the bidirectional synergy between understanding and generation in unified multimodal models, addressing a fundamental gap in current research.

2. The benchmark design is systematic and fine-grained, covering 10 task categories and 32 subtasks across two dimensions, UEG and GEU, and introducing both direct and step-by-step evaluation protocols for comprehensive diagnosis.

3. The experiments are extensive and carefully conducted, involving 12 unified and 6 task-specific models. The analysis offers valuable insights into the limitations of current unified models and the challenges of achieving genuine multimodal integration.

**Weaknesses:**

1. The dataset scale is relatively small, containing only 1,000 annotated samples, which may limit the benchmark’s ability to reflect the diversity and complexity of real-world multimodal interactions. Moreover, with 32 subtasks, this means each subtask has only about 30 samples on average, which is insufficient given the relatively coarse granularity of some categories, such as sports, making the distribution somewhat unbalanced and less representative.

2. As a comprehensive benchmark, RealUnify would benefit from broader task coverage similar to MMMU. The current task taxonomy lacks sufficient theoretical justification, and certain task definitions appear arbitrary. For example, the mathematical category in Figure 4 includes only four subtasks that do not fully represent mathematical reasoning, and the authors should clarify the design rationale behind such categorization.

3. The analysis of why unified models fail to achieve bidirectional synergy remains mostly descriptive and empirical. The paper lacks deeper theoretical discussion or modeling-based interpretation that could reveal the underlying mechanisms of such failures.

**Questions:**

Please refer to the weaknesses section, especially the w1 and w2.

---

### Official Review · Reviewer_Y3oM · 2025-11-03

**Soundness:** 3
**Presentation:** 2
**Contribution:** 2
**Rating:** 4
**Confidence:** 4

**Summary:**

This paper presents RealUnify, a benchmark designed to evaluate the visual understanding and generation capabilities of unified multimodal models. Experiments show that current models still lack the capability to synergize.

**Strengths:**

1. The proposed benchmark is comprehensive, covering diverse tasks and categories.
2. Part of the dataset is manually curated by humans.

**Weaknesses:**

1. The proposed benchmark has limited novelty compared to prior works. Both two settings RealUnify studied (i.e., knowledge/reasoning-enhanced image generation, and "think with image") has been widely studied by prior works. The interactions between understanding and evaluation has also been explored.

2. The authors' conclusion in lines 100-102 (i.e., decomposing GEU hurts performance because models rely on understanding shortcuts) is problematic. This performance degradation can be expected as image generation is intrinsically harder than understanding. More experiments are needed to rule out confounders before making such a conclusion.

3. The analysis and discussion between generation and understanding is limited. For example, the paper did not discuss the potential error propagation, and how the quality of intermediate images (either good or inferior), affects the understanding capability.

4. While discussing empirical findings, the authors did not provide actionable insights and promising pathways for future research for unified multimodal models.

5. The presentation in this paper is not clear and disorganized. For example, the critical details of the evaluation protocol and metrics are confusing. The "Stepwise Evaluation" paragraph in Section 3.3 simply re-explained the settings without explaining how stepwise evaluation is performed.

6. The evaluation is using LLM-as-a-Judge for several tasks, but this paper didn't discuss the reliability of this evaluation method on the evaluated dataset.

**Questions:**

1. In the UEG setting, how are the question lists generated? What measures have been taken to make sure the question list can provide a comprehensive evaluation of image quality? The design of question lists is focused on the accuracy of generation; what about the other aspects?

2. What is the data collection instruction and annotation criteria in Section 3.3? It's not clear what procedure has been taken to ensure the data quality.

3. What is the reliability of using Gemin-2.5-Pro as a judge?

---

### Note · Authors · 2025-11-13

I have read and agree with the venue's withdrawal policy on behalf of myself and my co-authors.